# Prior-Free Mechanism with Welfare Guarantees

## ABSTRACT

We consider the problem of designing prior-free revenue-maximizing mechanisms for allocating items to $n$ buyers when the mechanism is additionally provided with an estimate for the optimal welfare (which is guaranteed to be correct to within a multiplicative factor of $1/\alpha$). In the digital goods setting (where we can allocate items to an arbitrary subset of the buyers), we demonstrate a mechanism which achieves revenue that is $O(\log n/\alpha)$-competitive with the optimal welfare. In the public goods setting (where we either must allocate the item to all buyers or to no buyers), we demonstrate a mechanism which is $O(n \log 1/\alpha)$ competitive. In both settings, we show the dependence on $\alpha$ and $n$ is tight. Finally, we discuss generalizations to broader classes of allocation constraints.

## CCS CONCEPTS

• **Theory of computation → Algorithmic mechanism design**; **Computational pricing and auctions**.

## KEYWORDS

Prior-Free Mechanism Design, Digital Goods, Public Goods

**ACM Reference Format:**
. 2018. Prior-Free Mechanism with Welfare Guarantees. In *Proceedings of Make sure to enter the correct conference title from your rights confirmation emai (Conference acronym 'XX).* ACM, New York, NY, USA, 9 pages. https://doi.org/XXXXXXX.XXXXXXX

## 1 INTRODUCTION

One of the key assumptions made in the field of mechanism design is the nature and quantity of information the mechanism designer has regarding types/valuations of the agents in their problem. The study of various mechanism design problems has produced several popular models for this assumption.

In the standard economic approach of *Bayesian mechanism design*, these valuations are assumed to be drawn from distributions that are fully known to the mechanism designer. This enables the designer to tailor their mechanism to these prior distributions and facilitates the construction of robust revenue-optimizing mechanisms. For example, Myerson's auction [Mye81] is optimal for bidders whose values are drawn i.i.d. from a "regular" distribution. However, the real-world implementation of these mechanisms can be hindered due to the practical absence or inaccuracy of prior distributions. This research agenda to develop robust mechanisms that are less sensitive to modeling assumptions is often referred to as the "Wilson doctrine" [Wil85].

With this in mind, some more recent lines of work have tried to relax the assumption of access to a prior distribution. In *prior-independent mechanism design*, the mechanism designer knows that valuations are being drawn from some distribution but does not know the exact distribution. In this setting, assumptions like i.i.d. bidders and regular distributions can allow the designer to learn something about the overall distribution from a few samples.

Continuing to weaken our assumptions leads us to the regime of *prior-free mechanism design*, in which the mechanism designer knows nothing at all about the types of the agents. Mechanisms developed for this setting are generally simple and easier to practically apply, but are accompanied by the downside of having weaker guarantees. These guarantees generally take the form of competitive ratios to benchmarks that ignore the largest value of any bidder (e.g., [GH01] proves a constant-factor revenue competitive ratio with respect to $\max_{k \geq 2} k v_k$, where $v_k$ is the $k$th largest value). This is by necessity – knowing nothing at all about the valuations, it is impossible to get a constant factor approximation to the maximum possible revenue obtainable by an omniscient mechanism. But simultaneously, this is unsatisfying – there are many cases where one agent *could* be responsible for the bulk of the total value, and it would be ideal if we could have strong guarantees in these cases as well.

Very often it is the case that we are in an intermediate regime where, as a mechanism designer, we have some partial knowledge about the values of agents participating in our mechanism, but nothing approaching bidders drawing independently from value distributions that we can hope to learn enough about in the short-term. In this paper, we study the classical mechanism design problem of allocating identical items to unit-demand agents, augmented with an "estimate range" of the welfare of the best possible allocation (alternatively, the maximum revenue we could achieve if we precisely knew the valuation of each agent). For example, consider a setting where a corporation is trying to sell copies of a digital good to a population of buyers. By examining the earnings reports of other companies which have released similar products, the corporation might be able to estimate that the total value among all buyers of that good is within a certain range (e.g. "one million to two million dollars") but not have any other information about how that value is distributed among buyers (e.g. it all may come mostly from one buyer, or may be distributed equally among many buyers).

### 1.1 Our results

More formally, we consider the question where there are $n$ unit-demand buyers, and we are allowed to allocate (identical) items to some subset $S \in 2^{[n]}$ of the buyers, where $S$ must belong to a collection $\mathcal{S}$ of allowed subsets. Modifying $\mathcal{S}$ allows us to encode a variety of different mechanism design settings; our main settings of interest include:

- In the *digital goods* setting, $\mathcal{S} = 2^{[n]}$. This means we have an unlimited quantity of items to assign and may allocate to any subset of the buyers.

- In the *matroid* setting, $\mathcal{S}$ is a matroid. This means we can allocate the items to any independent set of the matroid. This is a generalization of the digital goods setting, and includes other interesting cases as well, e.g. if $\mathcal{S}$ is the $k$-uniform matroid, then that means we only have $k$ items to allocate to buyers but otherwise may do so freely.
- Finally, in the *public goods* setting, $\mathcal{S} = \{\emptyset, [n]\}$. This means if we allocate the item to any buyer, we must allocate it to all buyers.

Each buyer $i \in [n]$ has some value $v_i \in \mathbb{R}_+$ for being allocated an item. Our goal is to design a dominant-strategy truthful mechanism for this setting with good revenue guarantees. The mechanism does not have access to the individual values $v_i$, but does have some information about these valuations in the form of a hint about the total welfare $W = \max_{S \in \mathcal{S}} \sum_{i \in S} v_i$. Specifically, we assume that we know that $W$ lies in an interval $[L, R]$ with $L/R = \alpha > 0$. We say that a mechanism is $\beta$-*competitive* if the mechanism is guaranteed to generate at least $W/\beta$ revenue regardless of the specific values $v_i$.

We prove the following results:

- In the digital goods setting, there exists an $O(\log(n/\alpha))$-competitive mechanism. This is tight in both parameters: any $\beta$-competitive mechanism must satisfy $\beta = \Omega(\log n)$ and $\beta = \Omega(\log 1/\alpha)$.
- In the matroid setting, there exists an $O(\log(k/\alpha))$-competitive mechanism, where $k$ is the rank of the matroid. Similarly, this is tight in both $k$ and $\alpha$.
- Finally, in the public goods setting, there exists an $O(n \log(1/\alpha))$-competitive mechanism. Any $\beta$-competitive mechanism must satisfy $\beta = \Omega(n)$ and $\beta = \Omega(\log 1/\alpha)$.

All three mechanisms are efficient and straightforward to implement. The mechanism for the digital goods presents each buyer with a posted-price drawn randomly and independently from a specific distribution. The mechanism for the matroid setting extends the digital goods mechanism by first running a greedy algorithm to determine an eligible set of buyers and then subsequently running the digital goods mechanism on this subset. The public goods mechanism is the most unique of the three in both description and analysis – it involves choosing a threshold from a specific distribution (a mixture of several uniform distributions) and only allocating the item if the sum of the bids exceeds this threshold.

We conclude with a note on optimal mechanisms. When $n = 1$, our problem reduces to the problem of selling an item to a single buyer ($n = 1$) with unknown value in some range $[\alpha, 1]$. A classic result in mechanism design states that it is possible to get a $(1 + \log(1/\alpha))$-approximation of welfare as revenue, and that this approximation factor is tight (see e.g. chapter 7 of [Har11]; for convenience, we reproduce this result in Appendix A). Moreover, a very simple mechanism attains this approximation factor: randomly sampling a reserve from a specific "equal-revenue" distribution.

Inspired by this, one might hope that it is in fact possible to characterize the *optimal* mechanisms (with the best approximation factor) for more buyers given a range constraint on total welfare. Unfortunately, numerical simulations suggest that even for $n = 2$ buyers, the optimal mechanisms are quite bizarre. (We show how one can approximate the optimal mechanism in Appendix B.) In

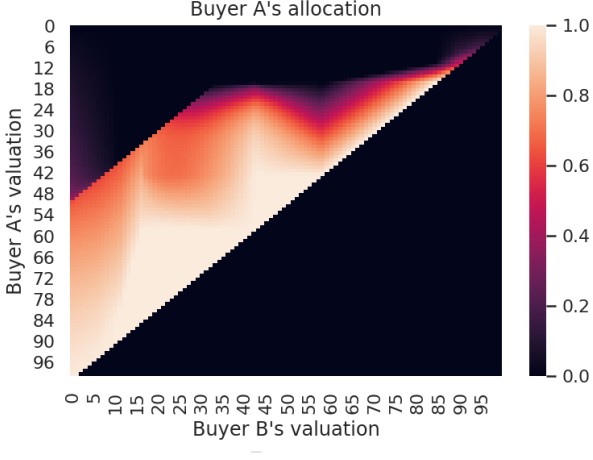

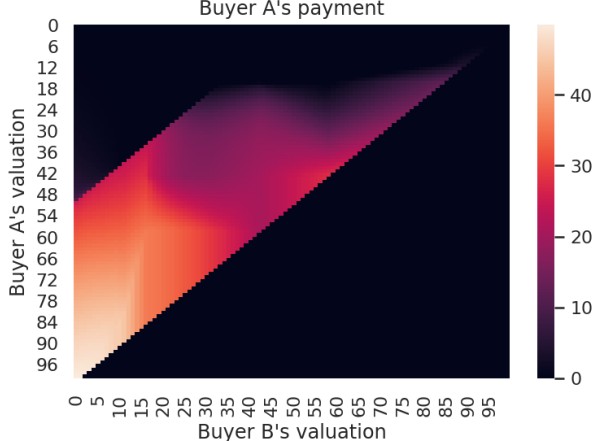

**Figure 1: Allocation / pricing rules for an approximately optimal mechanism for two buyers in the digital goods setting with welfare in $[50, 100]$ (we show only the allocation and payment of buyer A as a function of both buyers' valuations; buyer B is symmetric). This mechanism is $\beta$-competitive for $\beta \approx 0.486$. Appendix B explains how this figure was generated.**

Figure 1 we depict the allocation rule for the optimal mechanism in the digital goods setting with $n = 2$ and $\alpha = 1/2$. Note, for instance that the bidder's allocation is neither constant nor weakly monotone (in either direction) in the other bidder's value.

## 1.2 Related Work

There is a large body of work dedicated to revenue maximization on the spectrum between Bayesian mechanism design and completely prior-free mechanism design. We survey a couple of the more relevant strands here.

Our work most closely relates to the existing line of work on prior-free mechanism design. Early papers on prior-free mechanisms include [GH01, GHK+06], which provided worst-case revenue guarantees on selling digital goods. There has been a large amount of follow-up literature since then, tightening the known

competitive ratios [CGL14] and extending these mechanisms to broader settings [FGHK02, LR12]. All these papers work with absoutely no assumptions about the buyers' valuations, but in turn only provide revenue guarantees with respect to weaker benchmarks (e.g. "the revenue of the best fixed price less than the second-highest valuation").

Many lines of work study an intermediate regime where the designer is permitted some additional information about the buyers' types. One such approach towards auction design is that of distributionally-robust auctions. These auctions assume that the auctioneer has knowledge of some summary statistics of the distribution such as the mean and the upper limit of the support, and characterizes the max-min performance, i.e., under the worst case distribution (see [BTC22, Che22]. Another such approach focuses on the case where buyer valuations are drawn from a prior distribution but it is unknown and must be learnt (the "prior-independent setting"). [KL03] considered the case where one must learn the value of the buyers' distributions using posted-price mechanisms. [CR14] studied the question of determining how many samples one needs from a distribution to compute the optimal mechanism for revenue maximization. There is a line of work on approximately revenue-optimal auctions with access of 1 sample (see e.g. [DRY15, AKW14, CDFS19]).

The underlying mechanism design problem we study falls under the umbrella of "single-dimensional mechanism design" and has been extensively studied in Bayesian settings. In [KY13], the authors studied what revenue guarantees are possible under a variety of feasibility constraints (including the public goods and downwards closed settings) and regularity assumptions on the buyer distributions. [GNR18] proves computational hardness for the problem of designing revenue-optimal mechanism for public goods.

## 2 PRELIMINARIES

We build off the notation introduced in the introduction. For convenience, we will assume that the welfare $W$ lies in the interval $[\alpha, 1]$ (multiplicative rescaling of this range does not change any result). Our paper focuses on the regime where $\alpha$ is small and approaching zero, so we will assume throughout that $\alpha \in [0, \frac{1}{2}]$; the regime $\alpha \in [\frac{1}{2}, 1]$ is an interesting direction for future work.

*The bounded equal-revenue distribution.* In several of our lower bound proofs, we will need to make use of a specific bounded equal-revenue distribution. The equal-revenue distribution is a standard technique in algorithmic game theory; it is typically defined over the interval $[1, \infty)$ with the probability density function $f(z) = 1/z^2$ and has the property that if a buyer's value is being drawn from this distribution, any fixed price has revenue one in expectation:

$$\Pr_{z \sim \text{equal-revenue distribution}} [z \geq p] \cdot p = \int_p^\infty \frac{1}{z^2} dz \cdot p = \frac{1}{p} \cdot p = 1$$

Since we are interested in situations where the total welfare falls in the range $[\alpha, 1]$, it will be useful for us to employ equal-revenue distributions over bounded intervals $[\ell, h]$ where $0 < \ell \leq h$. We define these here and state their properties.

**Definition 2.1.** *Suppose we are given $\ell, h$ such that $0 < \ell \leq h$. Let the bounded equal-revenue distribution $\mathcal{D}_{eq}[\ell, h]$ be a probability distribution over $[\ell, h]$ that is $h$ with probability $\frac{\ell}{h}$ and is drawn from the distribution with probability density function $f(z) = \frac{h\ell}{(h-\ell)z^2}$ with probability $\frac{h-\ell}{h}$.*

**Lemma 2.2.** *Posting a price $p \in [\ell, h]$ against a buyer with value drawn from the bounded equal-revenue distribution $\mathcal{D}_{eq}[\ell, h]$ results in revenue $\ell$.*

**Lemma 2.3.** *The bounded equal-revenue distribution $\mathcal{D}_{eq}[\ell, h]$ has mean $\ell \left(1 + \log \frac{h}{\ell}\right)$.*

One way to think about the two previous lemmas is that they imply that if a mechanism designer got to see a valuation drawn from $\mathcal{D}_{eq}[\ell, h]$, they could get significantly more revenue (in fact, a multiplicative factor of $\left(1 + \log \frac{h}{\ell}\right)$ extra).

*The single-buyer setting.* In this section, we discuss the single-buyer case ($n = 1$). Note that in this case, our welfare information is actually just a bound on the value of this buyer, i.e. $v_1 \in [\alpha, 1]$. It is a folklore theorem that in this case, the best competitive ratio achievable is $\log(e/\alpha)$ (see e.g. chapter 7 of [Har11]). For completeness, we restate this here and include a proof in appendix A.

**Theorem 2.4.** *For the case of a single buyer, there exists a $\log(e/\alpha)$-competitive mechanism. Furthermore, this is tight; there is no $c$-competitive mechanism for any $c < \log(e/\alpha)$.*

## 3 DIGITAL GOODS SETTING

### 3.1 Upper Bound

We begin by presenting an $O(\log(n/\alpha))$-competitive mechanism for the digital goods setting. The mechanism itself is simple; we will run the single-bidder mechanism (from Theorem 2.4) on each buyer independently, under the assumption that their value lies in the interval $[\alpha/2n, 1]$ (i.e., with an $\alpha'$ of $\alpha/2n$). Since at least half of the welfare is contributed by bidders with value at least $\alpha/2n$, this leads to a $O(\log(1/(\alpha/2n))) = O(\log(n/\alpha))$ competitive mechanism. We prove this below.

**Theorem 3.1.** *The above mechanism is $2\log(2ne/\alpha)$-competitive.*

**Proof.** Note that if bidder $i$ has value $v_i$ at least $\alpha/2n$, by the guarantees of the single buyer mechanism (Theorem 2.4), this mechanism obtains revenue at least $v_i/\log(e/(\alpha/2n)) = v_i/\log(2ne/\alpha)$.

Now, let $W' = \sum_{i: v_i \geq \alpha/2n} v_i$ be the total welfare of all bidders with value at least $\alpha/2n$. As a consequence of the previous paragraph, our mechanism obtains revenue at least $W'/\log(2ne/\alpha)$. But note that we can write $W' = W - \sum_{i: v_i < \alpha/2n} v_i \geq W - \alpha/2 \geq W/2$ (where the last inequality follows since we are guaranteed that $W$ lies in the interval $[\alpha, 1]$). It follows that the above mechanism obtains revenue at least $W/(2\log(2ne/\alpha))$, as desired. □

### 3.2 Lower Bound

In this subsection, we prove the following lower bound that controls how the approximation ratio $\beta$ can scale as a function of the number of buyers $n$; in particular, we will show that the $\log n$ dependence on $n$ in the competitive ratio of Theorem 3.1 is necessary. Note that the $\log 1/\alpha$ dependence on $\alpha$ is necessary even in the $n = 1$ case by Theorem 2.4.

THEOREM 3.2. *Any $\beta$-competitive truthful mechanism for the digital goods setting with $n$ buyers and $\alpha = 1/2$ must satisfy $\beta = \Omega(\log n)$.*

PROOF. The plan is to consider a slightly-invalid distribution $\mathcal{D}$ whose support does not always have welfare in the range $[\frac{1}{2}, 1]$. We will show that (i) no mechanism can do well in an average-case sense on $\mathcal{D}$ and that (ii) if there is a mechanism $M$ that does well in a worst-case sense on $[\frac{1}{2}, 1]$, we can construct a mechanism $M'$ that does well in an average-case sense on $\mathcal{D}$. Together, this will show no mechanism can do well on $\mathcal{D}$.

Formally, $\mathcal{D}$ will be a product distribution with marginals equal to our bounded equal-revenue distribution $\mathcal{D}_{eq}[\ell, h]$. We now reason about how to choose $\ell$ and $h$. We will want to apply a Hoeffding bound on the sum of these bounded random variables to show that the total welfare is quite likely to be in the range $[\frac{1}{2}, 1]$. For the sum of $n$ independent random variables that are bounded between $[\ell, h]$ with expected sum $\mu$, for any $\delta > 0$ we have:

$$\Pr\left[\sum_i v_i \leq (1-\delta)\mu\right] \leq \exp\left(\frac{-2\delta^2\mu^2}{n(h-\ell)^2}\right)$$

$$\Pr\left[\sum_i v_i \geq (1+\delta)\mu\right] \leq \exp\left(\frac{-2\delta^2\mu^2}{n(h-\ell)^2}\right)$$

We would like to reason about the sum landing in the range $[\frac{1}{2}, 1]$, so we will aim to make $\mu = \frac{3}{4}$ and choose $\delta = \frac{1}{3}$. We will need the $(h-\ell)^2$ term in the bound to cancel out $n$ so $(h - \ell)$ should be roughly $\frac{1}{\sqrt{n}}$. With this in mind, we choose the following values.

$$\ell \triangleq \frac{3}{4n}\left(1 + \log\left(1 + \frac{1}{3}\sqrt{n}\right)\right)^{-1}$$

$$h \triangleq \left(\frac{3}{4n} + \frac{1}{4\sqrt{n}}\right)\left(1 + \log\left(1 + \frac{1}{3}\sqrt{n}\right)\right)^{-1}$$

We can now compute the values for $\mu$ and $(h - \ell)$. Invoking Lemma 2.3:

$$\mu = n \cdot \ell\left(1 + \log\frac{h}{\ell}\right)$$

$$= n \cdot \frac{3}{4n}\left(1 + \log\left(1 + \frac{1}{3}\sqrt{n}\right)\right)^{-1}\left(1 + \log\left(\frac{\frac{3}{4n} + \frac{1}{4\sqrt{n}}}{\frac{3}{4n}}\right)\right)$$

$$= \frac{3}{4}$$

$$h - \ell = \frac{1}{4\sqrt{n}}\left(1 + \log\left(1 + \frac{1}{3}\sqrt{n}\right)\right)^{-1}$$

Plugging this into our Hoeffding bound:

$$\exp\left(\frac{-2\delta^2\mu^2}{n(h-\ell)^2}\right)$$

$$= \exp\left(\left[-2\left(\frac{1}{3}\right)^2\left(\frac{3}{4}\right)^2\right] \Big/ \left[n\left(\frac{1}{4\sqrt{n}}\left(1 + \log\left(1 + \frac{1}{3}\sqrt{n}\right)\right)^{-1}\right)^2\right]\right)$$

$$= \exp\left(\left[-\frac{1}{8}\right] \Big/ \left[\left(\frac{1}{16}\left(1 + \log\left(1 + \frac{1}{3}\sqrt{n}\right)\right)^{-1}\right)^2\right]\right)$$

$$= \exp\left(-2\underbrace{\left(1 + \log\left(1 + \frac{1}{3}\sqrt{n}\right)\right)^2}_{\text{this is } \geq 1}\right)$$

$$\leq \exp(-2)$$

$$\leq 0.14$$

As a result, we know there is at most $2 \cdot 0.14 \leq 0.5$ probability that the sum is not in the range $[\alpha, 1]$ and hence at least $0.5$ probability that the sum is in this range.

With $\mathcal{D}$ now defined, we can reason about our first main claim, namely that no mechanism can do well on $\mathcal{D}$. For the sake of contradiction, suppose we have a mechanism $M'$ that could handle all inputs in the support of $\mathcal{D}$. Furthermore, suppose that for a random valuation drawn from $\mathcal{D}$, $M'$ achieves a $\beta'$ fraction the expected welfare of that valuation (note this is an average-case guarantee, not a worst-case guarantee). But we know the expected welfare of a valuation drawn from $\mathcal{D}$, so:

$$\text{Rev}(M', \mathcal{D}) \geq \frac{3}{4\beta'}$$

At the same time, $M'$ cannot do that well against a product distribution of bounded equal-revenue distributions. Because it is a product distribution, each valuation provides no information about the other valuations. Hence the expected revenue of our mechanism $M'$ cannot be better than just picking a fixed price for each bidder:

$$\text{Rev}(M', \mathcal{D}) \leq \frac{3}{4}\left(1 + \log\left(1 + \frac{1}{3}\sqrt{n}\right)\right)^{-1}$$

$$\beta' \geq \left(1 + \log\left(1 + \frac{1}{3}\sqrt{n}\right)\right)$$

This completes our first main claim; we know that mechanisms can only do so well on $\mathcal{D}$. We now prove our second main claim, namely that a good mechanism $M$ on the interval $[\frac{1}{2}, 1]$ implies a good mechanism $M'$ on $\mathcal{D}$. Suppose we have a $\beta$-competitive mechanism $M$ over valuations that have welfare in $[\frac{1}{2}, 1]$. We will extend it to a mechanism $M'$ over valuations in the support of $\mathcal{D}$.

We do so by reasoning about allocation functions, from which the pricing function (to make the entire mechanism truthful) can be recovered via Myerson's Lemma [Mye81]. The allocation function of $M'$ is as follows:

- If the input $v$ has welfare less than $\frac{1}{2}$, all buyers receive nothing.
- If the input $v$ has welfare in the range $[\frac{1}{2}, 1]$, all buyers receive the same as they would under $M$.

- If the input $v$ has welfare greater than 1, then each buyer receives the good.

Observe that our new allocation function is monotone as well, since increasing a buyer's valuation can only bring welfare from less than $\frac{1}{2}$ to in the range $[\frac{1}{2}, 1]$ to greater than one, which increases what the buyer gets from nothing to the original valuation function to one. When we apply Myerson's Lemma to this allocation function, we essentially get the same payment rule as the original mechanism $M$ had, but extended to be zero for inputs $v$ with welfare less than $\frac{1}{2}$ and some nonnegative amount for inputs with welfare greater than one. In other words, we know that $M'$ will obtain at least a $\beta$-fraction of the welfare as revenue when the welfare is in the range $[\frac{1}{2}, 1]$ and a nonnegative amount otherwise. Worst-case, we know that $M'$ achieves expected revenue at least $0.25\beta$.

We have already computed that this distribution has expected welfare $\frac{3}{4}$, so we can compute an average-case guarantee of $M'$ on $\mathcal{D}$:

$$\text{Rev}(M', \mathcal{D}) \geq 0.25/\beta$$
$$= \left(\frac{1}{3\beta}\right)\frac{3}{4}$$

Hence on $\mathcal{D}$, $M'$ achieves a $\beta' = 3\beta$ fraction of the welfare as revenue in expectation (an average-case guarantee). We can now combine this with our first main claim.

$$\beta \geq 3\left(1 + \log\left(1 + \frac{1}{3}\sqrt{n}\right)\right)$$
$$\geq 3\log\frac{1}{3}\sqrt{n} = \Omega(\log n).$$

This completes the proof. □

**Remark 3.3.** *For any fixed $\alpha < 1$, it is possible to adapt the proof of Theorem 3.2 to show that for sufficiently large $n$, any mechanism for the digital goods setting must be $\Omega(\log n)$ competitive. However, as $\alpha$ gets closer to 1, the constant in this $\Omega(\log n)$ worsens (since it becomes increasingly hard to ensure the sum of values lands in the range $[\alpha, 1]$ with high probability). It is an interesting open question to understand the dependence of the optimal competitive ratio as both $\alpha$ approaches 1 and as $n$ goes to infinity (i.e., the regime where our estimate for welfare is very accurate, but this welfare is divided among many bidders).*

## 4 MATROID SETTING

In this section, we generalize the results from our digital goods setting to the matroid setting.

### 4.1 Upper Bound

We present a $O(\log(k/\alpha))$-competitive for the matroid setting (where $k$ is the rank of the matroid). We use a two-phase mechanism:

(1) In the first phase, we sort all all bidders in order of non-increasing bid (breaking ties arbitrarily but consistently). Initially, the set of bidders we will pass to the next phase is an empty set. When we process a bidder, we add it to the set that we will pass if doing so produces an independent set.

(2) In the second phase, we receive an independent set and run the digital goods mechanism (from Theorem 3.1) on this set

of $k$ buyers under the assumption that the total welfare is in $[\alpha, 1]$.

**THEOREM 4.1.** *The above mechanism is incentive-compatible, produces an independent set, and is $2\log(2ke/\alpha)$-competitive.*

**PROOF.** The plan is to analyze the proposed mechanism using Myerson's Lemma [Mye81]. Note that we have only specified an allocation rule, but Myerson's Lemma describes how to construct a corresponding payment rule to make an overall truthful mechanism as long as our allocation rule was monotone.

To do so, it will be helpful to introduce some notation to characterize the effects of the two phases. Consider some buyer $i$, and fix the bids of all other buyers to be some $b_{-i}$. In order to get past phase one, this buyer must be early enough in the greedy ordering such that adding buyer $i$ still yields an independent set. Hence there is some minimum bid $\tau(b_{-i})$ (technically an infimum due to tiebreaking issues) they must present to pass the first phase. Then, assuming they passed this check, $b_{-i}$ uniquely determines the (up to) $k - 1$ other bidders that also passed. As a thought experiment, suppose buyer $i$ was allowed to skip the first phase and enter the second phase with these $k - 1$ other bidders determined by $b_{-i}$. Then they would face just the digital goods mechanism of Theorem 3.1, which results in some allocation rule for bidder $i$: $\tilde{x}_{b_{-i}}(b_i)$. Using these definitions, the overall allocation rule imposed on buyer $i$ is:

$$x_{b_{-i}}(b_i) = \begin{cases} 0 & \text{if } b_i < \tau(b_{-i}) \\ \tilde{x}_{b_{-i}}(b_i) & \text{otherwise} \end{cases}$$

This rule is monotone because it is a truncated version of the allocation rule for the digital goods mechanism, which is monotone due to that mechanism being truthful (Myerson's lemma states monotonicity and implementability for allocation rules are equivalent).

Next, it is simple to confirm that our mechanism always produces an independent set because the first phase produces an independent set by construction and the second phase always produces a subset of that (subsets of independent sets are independent).

It just remains to prove that the mechanism is $2\log(2ke/\alpha)$-competitive. To do so, we will want to think about the payments induced by our allocation rule. We now consider some buyer $i$ in the optimal independent set, i.e. the one which achieves optimal welfare $W \in [\alpha, 1]$. This is the same set that the greedy algorithm in phase one produces, since the greedy algorithm is optimal over matroids [Edm71]. What payment rule does this buyer face? We know that if they only went through phase two, we can apply Myerson's Lema to generate a payment rule from the allocation rule $\tilde{x}_{b_{-i}}(b_i)$. Instead, we generate a payment rule from the truncated version $x_{b_{-i}}(b_i)$. Relative to $\tilde{x}$, our allocation rule $x$ winds up delaying any allocation increases until at least $\tau(b_{-i})$. As a result, for buyer values which are at least $b_{-i}$, we extract at least as much payment as the corresponding payment rule for $\tilde{x}$. But we know that buyer $i$ bids truthfully and their value is at least $\tau(b_{-i})$ since $i$ is in the optimal independent set by assumption.

Hence for all buyers in the optimal independent set, we extract as least as much payment as directly running the digital goods mechanism (i.e. only phase two) on these buyers. But Theorem 3.1 guarantees that would extract a $1/(2\log(2ke/\alpha))$-fraction of the

welfare of these buyers. This proves our approximation guarantee and completes the proof. □

**Remark 4.2.** *It is possible to generalize the mechanism and proof in this section to work for a downward-closed set $\mathcal{S}$, with the caveats that (i) we lose efficiency from the first phase of the mechanism, which needs to optimize over $\mathcal{S}$ and (ii) the exact guarantee becomes more complicated because it depends on the size of the set that enters the second phase of the mechanism (note it is not necessarily true that all maximal sets in $\mathcal{S}$ have the same cardinality).*

## 4.2 Lower Bound

We also inherit our lower bound from the digital goods setting; this proof is much more straightforward. Interestingly, our lower bound does not simply show some matroid of rank $k$ is hard, but rather every matroid of rank $k$ must be hard.

**Corollary 4.3.** *Fix any matroid $\mathcal{M}$ over $n$ elements with rank $k$. Any $\beta$-competitive truthful mechanism for the matroid setting with $\mathcal{M}$ and $\alpha = 1/2$ must satisfy $\beta = \Omega(\log k)$.*

Proof. Since $\mathcal{M}$ is rank $k$, it must have an independent set of size $k$. We invoke Theorem 3.2 to produce a distribution over that set of buyers, then pad out to the $n$ total buyers by adding $(n - k)$ buyers who have value zero. Observe that this maintains the welfare $W$. Then any hypothetical $\beta$-competitive truthful mechanism for $\mathcal{M}$ and $\alpha = 1/2$ extracts too much revenue from the real $k$ buyers (it cannot extract any payments from the zero-value buyers we used to pad), which violates Theorem 3.2. This completes the proof. □

## 5 PUBLIC GOODS SETTING

### 5.1 Upper Bound

As with digital goods, we begin this section by presenting a mechanism for the public goods setting. The mechanism we present will be $O(n \log(1/\alpha))$-competitive with the welfare; later we will show this has the optimal dependence on $n$ and $\alpha$.

We first remark that one natural approach for designing such a mechanism is to attempt to mirror the approach in Theorem 3.1 and run some instance of the single-buyer mechanism per bidder. In the public goods setting, we can accomplish this as follows: divide the item into $n$ equal pieces, and sell the $i$th piece independently to bidder $i$ via the single-buyer mechanism (with the understanding that if bidder $i$ wins this piece, all other bidders also receive it). Doing this results in an $O(n \log(n/\alpha))$-competitive mechanism, but it turns out this dependence on $n$ is sub-optimal.

Instead, we will show that we can do asymptotically better by using a *threshold mechanism*. Specifically, we will sample a threshold $\tau$ from some distribution depending on $\alpha$ (defined in Mechanism 1) and allocate the item only if the sum of values $W$ (truthfully reported as bids) is at least $\tau$. If allocation occurs, we charge bidder $i$ a fee of $\max(0, \tau - (W - v_i))$ (the VCG payment rule for this mechanism). We show this mechanism is $O(n \log(1/\alpha))$-competitive (improving over the best posted-price algorithm by an $O(\log n)$ factor).

**Theorem 5.1.** *Mechanism 1 is truthful and $2ne \log(1/\alpha)$-competitive.*

Proof. To see that Mechanism 1 is truthful, note that each bidder pays the minimum amount they would have to bid to cause the

---

**Algorithm 1:** An $O(n \log(1/\alpha))$-competitive mechanism for the public goods setting.

**Input:** Welfare estimate $[\alpha, 1]$, $n$ bidders (with private values $v_i$).

Sample $\omega$ uniformly from the set $\mathcal{W} = \{\alpha, e\alpha, e^2\alpha, \ldots, 1\}$.

Sample $\tau$ uniformly from the interval $[0, \omega]$.

Solicit (truthfully) bids $b_i$ from each bidder.

Allocate the item only if $\sum_i b_i \geq \tau$. In this case, charge bidder $i$ a cost of $r_i = \max(0, \tau - \sum_{j \neq i} b_j)$.

---

item to be (deterministically) allocated, so by Myerson's Lemma this payment rule is truthful (and the bid $b_i$ submitted by bidder $i$ will be $v_i$). We now proceed to analyze the competitive ratio of this mechanism.

Note that by the construction of the set $\mathcal{W}$, there is an element $\omega$ in $W$ with the property that $W \leq \omega \leq eW$. Since we have a $1/\log(1/\alpha)$ probability of choosing this element $\omega$ from $\mathcal{W}$, let us condition on this event (this adds a $|\mathcal{W}| = \log(1/\alpha)$ factor to our eventual competitive ratio).

Now, consider the revenue $r_i = \max(0, \tau - (W - v_i))$ we obtain from bidder $i$ through this mechanism. Since $\tau$ is chosen uniformly in $[0, \omega]$, this is a random variable with expectation

$$\mathbb{E}[r_i] = \frac{1}{\omega} \int_{W-v_i}^{\omega} (\tau - (W - v_i)) d\tau = \frac{1}{2\omega}(\omega - W + v_i)^2 \geq \frac{v_i^2}{2\omega}.$$

This implies the total expected revenue from the mechanism (conditioned on choosing the correct $\omega$) is at least

$$\sum_{i=1}^{n} \mathbb{E}[r_i] \geq \frac{1}{2\omega} \sum_{i=1}^{n} v_i^2 \geq \frac{1}{2n\omega} \left( \sum_{i=1}^{n} v_i \right)^2 = \frac{W^2}{2n\omega} \geq \frac{1}{2ne} W.$$

□

### 5.2 Lower Bound

We now show the linear dependence on $n$ is tight (as with digital goods, the $\log(1/\alpha)$ dependence on $\alpha$ is tight even in the $n = 1$ case by Theorem 2.4).

**Theorem 5.2.** *Any $\beta$-competitive truthful mechanism for the public goods setting with $n$ buyers and $\alpha = 1/2$ must satisfy $\beta = \Omega(n)$.*

Proof. For simplicity, we prove the claim for even $n$ (odd $n$ can be handled by padding the even case with an additional bidder that always has zero valuation).

Our counterexample valuation vectors for $n$ bidders will all follow the restriction that each bidder's valuation is either zero or $\frac{1}{n}$; due to $\alpha$, at least $\frac{n}{2}$ of the bidders must have the latter value. The performance of any mechanism on these valuation vectors can be characterized by its allocation function on these inputs. Although it initially appears that the space of possible mechanisms is large, we claim that a mechanism might as well give the same allocation to any permutation of a valuation vector.

Formally, suppose we have a candidate mechanism with allocation function $x(v)$. Consider an symmetrized version with allocation

function $\tilde{x}(v)$ defined by:

$$\tilde{x}(v) \triangleq \frac{1}{n!} \sum_{\sigma \in S_n} x(\sigma(v))$$

The payment rule works out to:

$$
\begin{aligned}
\tilde{p}(v) &= \sum_{i=1}^{n} v_i \cdot (\tilde{x}(v) - \tilde{x}(0, v_{-i})) \\
&= \sum_{i=1}^{n} v_i \cdot \left( \frac{1}{n!} \sum_{\sigma \in S_n} x(\sigma(v)) - x(\sigma(0, v_{-i})) \right) \\
&= \frac{1}{n!} \sum_{\sigma \in S_n} \sum_{i=1}^{n} v_i \cdot (x(\sigma(v)) - x(\sigma(0, v_{-i}))) \\
&= \frac{1}{n!} \sum_{\sigma \in S_n} p(\sigma(v))
\end{aligned}
$$

But this implies that the worst point of our symmetrized mechanism does at least as well as the worst point of the original mechanism:

$$\tilde{p}(v) \geq \min_{\sigma \in S_n} p(\sigma(v))$$

$$\frac{\tilde{p}(v)}{\|v\|} \geq \min_{\sigma \in S_n} \frac{p(\sigma(v))}{\|\sigma(v)\|}$$

Hence it suffices to only consider symmetrized mechanisms when proving our counterexample. Since all our bidders only have one of two values and we have symmetrized, the allocation function might as well only depend on the total welfare $W = \|v\|$. Consider the allocations when the total welfare is $W = \frac{1}{2}, \frac{1}{2} + \frac{1}{n}, \cdots, 1$. We know that:

$$0 \leq x\left(\frac{1}{2}\right) \leq x\left(\frac{1}{2} + \frac{1}{n}\right) \leq \cdots \leq x(1) \leq 1$$

By the pigeonhole principle, we know that exists an $i$ between $\frac{n}{2}$ and $n$ such that the difference $x(\frac{i+1}{n}) - x(\frac{i}{n})$ is at most $\frac{2}{n}$. But this implies that:

$$
\begin{aligned}
p\left(\frac{i+1}{n}\right) &= \frac{i+1}{n} \cdot \left( x\left(\frac{i+1}{n}\right) - x\left(\frac{i}{n}\right) \right) \\
&\leq W \cdot \frac{2}{n}
\end{aligned}
$$

This completes the proof. □

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

## A OMITTED PROOFS

*Proof of Lemma 2.2.*

PROOF OF LEMMA 2.2. This is straightforward to verify, keeping in mind that drawing a value of $h$ is always larger than $p \in [\ell, h]$.

$$\Pr_{z \sim \mathcal{D}_{eq}[\ell,h]}[z \geq p] \cdot p = \left[ \int_p^h \frac{\ell}{z^2} dz + \frac{\ell}{h} \right] \cdot p$$
$$= \left[ \frac{\ell}{p} - \frac{\ell}{h} + \frac{\ell}{h} \right] \cdot p$$
$$= \ell$$

This completes the proof. □

*Proof of Lemma 2.3.*

PROOF OF LEMMA 2.3. We have:

$$\mathbb{E}_{z \sim \mathcal{D}_{eq}[\ell,h]}[z] = \int_\ell^h \frac{\ell}{z} dz + \frac{\ell}{h} \cdot h$$
$$= \ell \log \frac{h}{\ell} + \ell$$
$$= \ell \left( 1 + \log \frac{h}{\ell} \right)$$

□

PROOF OF THEOREM 2.4. Suppose that there is a truthful mechanism that achieves an $\lambda$-approximate mechanism. Let $x(v)$ denote the probability of receiving an item when the bid is $v$. WLOG, we assume $x(v)$ is nonzero iff $v \in [\alpha, 1]$. From Myerson's Lemma [Mye81], we know that the allocation rule $x(v)$ is implementable if and only if it is monotone, and if it is implementable the corresponding payment rule is $p(v) = \int_\alpha^v t \cdot x'(t) dt$. Observe that the optimal allocation rule $x$, will satisfy the following constraints:

$$\int_\alpha^v t \cdot x'(t) dt \geq \lambda v \qquad \forall v \in [\alpha, 1] \qquad (\text{LP}_\epsilon)$$
$$\int_\alpha^1 x'(t) \leq 1$$
$$x'(t) \geq 0 \qquad \forall t \in [\alpha, 1]$$

We will show that we can construct a sequence of linear programs $LP_\epsilon$ such that they can approximate the optimal allocation upto a discretization error of $\epsilon$. We show that these linear programs yield an allocation rule whose approximation ratio that converges to a value of $\frac{1}{1-\log(\alpha)}$.

Let $\mathcal{I}_\epsilon$ be the set that includes the multiples of $\epsilon$ along with $\alpha$, i.e. $\mathcal{I}_\epsilon = \{0, \epsilon, 2\epsilon, \dots 1\} \cup \{\alpha\}$ denote the discretization of interval $[0, 1]$ into multiples of $1/\epsilon$ along with $\alpha$. Denote by $suc(x) = \min_{y \in \mathcal{I}_\epsilon, y > x} y$ and $pred(x) = \max_{y \in \mathcal{I}_\epsilon, y < x} y$ to be the successor for a given value $x$ and predecessor of a given value $x$ in the discretization $\mathcal{I}_\epsilon$.

For any $\epsilon$, we can show that there exists a family of linear programs $LP_\epsilon$ parametrized by $\epsilon$ which upper bound the optimal approximation factor $\lambda$ by at most $(1 + \epsilon)$. Taking the limit $\epsilon \to 0$, we

obtain the desired result. Consider the following linear program

$$\max \lambda \qquad (\text{Primal}_\epsilon)$$
$$\sum_{t \in \mathcal{I}_\epsilon, t \leq v} (t + \epsilon) \cdot y_t \geq \lambda v \qquad \forall v \in \mathcal{I}_\epsilon \cap [\alpha, 1] \qquad (\text{apx})$$
$$\sum_{t \in \mathcal{I}_\epsilon} y_t \leq 1 \qquad (\text{alloc})$$
$$y_t \geq 0 \qquad \forall t \in \mathcal{I}_\epsilon \qquad (\text{mon})$$

First observe that given an allocation function $x$ with a approximation value, we can set $y_t = x(suc(t)) - x(t)$ and $\lambda$ to be the same as the approximation value. Note that eq. (alloc) and eq. (mon) are immidiately satisfied due to the monotonicity and the definition of an allocation function. The eq. (apx) will translate to $\sum_t (t + \epsilon) \cdot (x(suc(t)) - x(t)) = \sum_t (t + \epsilon) \cdot \int_t^{t+\epsilon} x(suc(t)) - x(t) \geq \int_\alpha^v t \cdot x'(t) dt \geq \lambda v$.

To show that the optimal value of $\text{LP}_\epsilon$ lies in $\left[ \frac{1}{1-\log(\alpha)}, \frac{1+\epsilon}{1-\log(\alpha)} \right]$, we will present a primal solution with value $\frac{1}{1-\log(\alpha)}$ and a dual solution whose value is at most $\frac{(1+\epsilon)}{1-\log(\alpha)}$. and we set $y_t = f(suc(t)) - f(t)$ where $f$ is the function defined below:

$$f(t) = \begin{cases} \frac{1}{1-\log(\alpha)} \left( 1 + \log(t/\alpha) \right) & \text{if } t \in [\alpha, 1] \\ 0 & \text{else} \end{cases}$$

Note that this function has the derivative

$$f'(t) = \begin{cases} \frac{1}{1-\log(\alpha)} \cdot \frac{1}{t} & \text{if } t \in [\alpha, 1] \\ 0 & \text{else} \end{cases}$$

.

To show that this is tight, we exhibit the dual:

$$\min \beta \qquad (\text{Dual}_\epsilon)$$
$$\beta - (t + \epsilon) \cdot \sum_{v: t \leq v \leq 1} z_v \geq 0 \qquad t \in \mathcal{I}_\epsilon$$
$$\sum_{v \in \mathcal{I}_\epsilon} v \cdot z_v \geq 1$$
$$z_v \geq 0 \qquad \forall v \in \mathcal{I}_\epsilon$$
$$\beta \geq 0$$

The above dual has the following solution: $\beta = \frac{1+\epsilon}{1-\log(\alpha)}$ and

$$g(v) = \begin{cases} \frac{1}{1-\log(\alpha)} \cdot \left( 1 - \frac{1}{v} \right) & \text{if } v \in [\alpha, 1] \\ 0 & \text{else} \end{cases}$$

and set $z_v = g(v) - g(pred(v))$. By construction, we notice that $g(1) - g(t) = \frac{1}{1-\log(\alpha)} \frac{1}{t}$. The first constraint is satisfied as $\frac{1+\epsilon}{1-\log(\alpha)} - (t + \epsilon) \cdot \left( \frac{1}{1-\log(\alpha)} \frac{1}{t} \right) \geq 0$.

The second constraint is satisfied as

$$\sum_{v \in \mathcal{I}_\varepsilon} v z_v$$

$$= \sum_{v \in \mathcal{I}_\varepsilon} v \cdot (g(v) - g(\text{pred}(v)))$$

$$= g(1) - \alpha \cdot g(\alpha) + \varepsilon \cdot \sum_{v \in \mathcal{I}_\varepsilon, \alpha \le v \le 1} g(v)$$

$$\ge g(1) - \alpha \cdot g(\alpha) + \int_\alpha^1 g(v) dv$$

$$= 1$$

$\square$

# B COMPUTING APPROXIMATELY OPTIMAL MECHANISMS FOR TWO (OR MORE) BUYERS

In this appendix, we explain how to generalize the linear program in the proof of Theorem 2.4 to compute approximately optimal mechanisms for two buyers (which we used to generate Figure 1).

A useful observation is that without loss of generality, the optimal mechanism is symmetric in the two bidders. This is because any $\beta$-competitive truthful asymmetric mechanism can be converted into a $\beta$-competitive truthful symmetric mechanism by averaging it with a version that has the bidders swapped[1] due to linearity. Hence we only need a linear program to find the best symmetric mechanism.

This assumption lets us just consider a one-dimensional allocation function $x(v_1, v_2)$ which represents the allocation to buyer one when they bid $v_1$ and the other buyer bids $v_2$; symmetry allows us to retrieve buyer two's allocation function by swapping the arguments: $x(v_2, v_1)$. Again, from Myerson's Lemma [Mye81] we know that this allocation rule is implementable if and only if it is monotone, and if it is implementable the corresponding payment rule is $p(v_1, v_2) = \int_0^{v_1} t \cdot x'(t, v_2) dt$. Applying the same discretization as before (recall that $\mathcal{I}_\varepsilon = \{0, \varepsilon, 2\varepsilon, ..., 1\} \cup \{\alpha\}$) yields this approximate linear program:

$$\max \lambda$$

$$p_{v_1, v_2} = \sum_{t \le v_1} (t + \epsilon) \cdot y_{t, v_2} \qquad \forall v_1, v_2 \in \mathcal{I}_\varepsilon$$

$$p_{v_1, v_2} + p_{v_2, v_1} \ge \lambda(v_1 + v_2) \qquad \forall v_1, v_2 \in \mathcal{I}_\varepsilon \text{ s.t. } v_1 + v_2 \in [\alpha, 1]$$

$$\sum_{t \in \mathcal{I}_\varepsilon} y_{t, t'} \le 1 \qquad \forall t' \in \mathcal{I}_\varepsilon$$

$$y_{t, t'} \ge 0 \qquad \forall t, t' \in \mathcal{I}_\varepsilon$$

As before, when we take the limit $\varepsilon \to 0$ this linear program approaches the optimal mechanism.

Received 20 February 2007; revised 12 March 2009; accepted 5 June 2009

---

[1]In general, mechanisms for symmetric settings with more than two bidders can also be symmetrized by averaging them over all permutations of bidders.

