# OpenReview forum: "Prior-Free Mechanisms with Welfare Estimates"
_ACM.org/TheWebConf/2024/Conference — TheWebConf24_

### Official Review · Reviewer_N8JH · 2023-11-19

**Novelty:** 5
**Technical Quality:** 6

**Review:**

### Summary

The paper studies prior-free mechanism design where the goal is to maximize revenue as a fraction of the optimal welfare in the worst case.  The authors consider single-parameter buyers and binary allocations.  The main results are tight approximation bounds for (1) the matroid setting, (2) by implication, the digital good setting, and (3) the public good setting where either all buyers are allocated or none.


### Strengths

The paper studies a clean and meaningful problem and provides complete and tight results.  The proofs are not super lengthy but exhibit some interesting ideas.  A minor point is that the proofs are very well written and easy to follow, and I appreciate the authors' candidness not trying to make things look more complicate than they are.


### Weaknesses

I'd be more excited to also see bounds for "revenue vs revenue", which would capture other, equally important aspects of the hardness of these settings.

**Questions:**

Lines 137-144: technically, isn't the first bullet point a special case of the second?

Lower bounds: it seems these bounds are for "revenue against welfare" (which is reasonable and matches the upper bounds in the paper).  Still I wonder if it's possible to get better bounds if one uses the optimal BIC revenue as the benchmark.

Properties of equal revenue distributions: the authors already say this, but I'd cite something to make it clear these were known before (it doesn't hurt anyways).

Remart 4.2, point (i): without thinking too much, it seems one can just replace phase (1) of the matroid mechanism by picking the set of bidders with the largest total value (and is feasible)?  This is monotone in each v_i and doesn't seem to lose efficiency?  Then the approximation ratio would depend on the rank of the feasibility constraint, which is the size largest feasible set (this is also a fairly common parameter especially when downward-closed feasibility is under consideration)?

**Reviewer Confidence:**

3: The reviewer is confident but not certain that the evaluation is correct

**Scope:**

4: The work is relevant to the Web and to the track, and is of broad interest to the community

---

### Official Review · Reviewer_W93Z · 2023-11-21

**Novelty:** 5
**Technical Quality:** 4

**Review:**

This paper studies the problem of designing prior-free revenue-maximizing mechanisms for allocating identical items to buyers. They focus on the scenario where the mechanism is provided with an estimate for the optimal welfare, which is guaranteed to be correct within a certain ratio. This paper presents mechanisms for the digital goods setting (with no constraint on the set of allocated buyers), the matroid setting (with the constraint that the set of allocated buyers form an independent set), and the public goods setting (with the constraint of sell-to-all or sell-to-no-one) and analyses their competitive ratios. The mechanisms for digital goods setting and the matroid setting are designed based on the single-buyer mechanism. For public goods setting, this paper gives a threshold mechanism. For all three settings, they show that the mechanisms achieve (asymptotically) optimal approximation ratios.

The model of prior-free revenue-maximizing mechanisms is interesting and the results of tight ratio are strong. However, the techniques used to derive these results seem quite simple. The writing is the paper is in general good but not very easy to read to non-experts. Also, this paper is very short. I think the authors should use more words to explain the basic concepts, notions, and definitions (to non-experts). For example, how does a “mechanism” work, what are the actions of buyers and sellers, what does truthful mean, etc. The uses of terminologies are sometimes arbitrary: please formally define what are “posted price”, “revenue”, “welfare”, “buyer”, “bidder”, “value”, etc, preferably using mathematical notions.

Minor comments:
1.	Abstract, “a mechanism which achieves” should be “a mechanism that achieves”
2.	Line 515, “all all” should be “all”.
3.	Line 568, “Lema” should be “Lemma”.

**Questions:**

In Algorithm 1, what does e mean?

**Reviewer Confidence:**

2: The reviewer is willing to defend the evaluation, but it is likely that the reviewer did not understand parts of the paper

**Scope:**

3: The work is somewhat relevant to the Web and to the track, and is of narrow interest to a sub-community

---

### Official Review · Reviewer_Sx4J · 2023-11-21

**Novelty:** 4
**Technical Quality:** 5

**Review:**

This paper considers the question of designing auctions which obtain high revenue in a prior-free setting when the auctioneer has some (reasonably accurate) estimate of the optimal social welfare.  More concretely, the authors consider a setting in which an auctioneer faces a subset of bidders with private values who are looking to procure some good/service and the auctioneer knows that the optimal social welfare lies between a given “hint” $\alpha$ and $1$ (precisely, the authors consider a special case where $\alpha \in (0, 1/2]$.  The goal of the auctioneer is to maximize her expected revenue and she compares her revenue to the optimal social welfare, i.e., the maximum possible revenue collectable by any IC/IR mechanism ex-post.  In particular, the authors examine auctions with matroid settings, wherein subsets of bidders which can be feasibly served correspond to independent sets in an underlying matroid (e.g., digital goods settings), and a public goods setting, wherein the feasible subsets of bidders are the set of all bidders or the empty set.  In both settings, the authors give (asymptotically) tight approximation guarantees in terms of $n$ (the number of bidders) and $\alpha$ by providing mechanisms with upper bound performance guarantees and instances showing matching lower bounds against any mechanism.

On the positive side, the mechanisms proposed by the authors are simple and natural, yet optimal.  Moreover, the analysis is clearly presented and straightforward.  Finally, I appreciate that the authors examine two well-motivated and well-studied settings – the natural setting of matroid constraints over bidders (which is well studied economically) and the digital goods setting, which largely inspired the line of literature on prior-free auctions for revenue.  On the other hand, from a practical perspective it isn’t clear that the model of having an estimate of the optimal welfare but not other bidder level statistics (or aggregate estimates of distributional information) is natural, so, in my view, this paper is more theoretical than practical. Secondly, the paper only answers the question for $\alpha \leq 1/2$.  It would be interesting to give some discussion regarding challenges in handling the cases where $\alpha$ is large.  Finally, the techniques to prove the bounds and the mechanisms themselves are not too novel or surprising and it does not seem likely, in my view, that the results would be of significantly broad interest nor would the techniques lead to many new results elsewhere.  In summary, while there are positive aspects of this paper, there are also negative aspects which detract from the overall picture.  I outline some smaller comments below.

Line 132:  $L/R$ is a fraction less than $1$, you probably want to say $\alpha \leq 1$ here.

Line 276:  I wouldn’t call equal-revenue distributions “a technique”.  Instead, they are just a class of distributions.

Line 515: “all all” -> “all”

Line 584: Typically one refers to a set system as being downward-closed rather than just a set.

Line 639:  You refer to Algorithm 1 as Mechanism 1 in the text.  I would suggest following a consistent naming convention.

Line 730: “that exists” -> “that there exists”

[After rebuttal]  I thank the authors for their responses to my questions and the questions of the other reviewers.

**Questions:**

Can you comment on what settings we may expect an estimate of welfare to exist, but not other statistics common in the literature?

Can you comment on the case of $\alpha > 1/2$?

**Reviewer Confidence:**

3: The reviewer is confident but not certain that the evaluation is correct

**Scope:**

3: The work is somewhat relevant to the Web and to the track, and is of narrow interest to a sub-community

---

### Official Review · Reviewer_r88P · 2023-11-23

**Novelty:** 5
**Technical Quality:** 6

**Review:**

This paper studies a problem in which there are $n$ unit-demand buyers, and the goal is to design a dominant-strategy truthful mechanism to allocate identical items to the buyers with good revenue guarantees. The paper studies three different settings: i) digital goods: we can allocate to any subset of $n$ buyers, ii) matroid: the chosen subset of buyers should be an independent set of a matroid, and iii) public goods: the set of chosen buyers is either the null set or all the buyers. For each setting, the authors first provide a mechanism along with its competitive ratio guarantees, and then, they provide matching lower bounds to prove the optimality of their mechanisms.

The paper contributes to prior-free single-dimensional mechanism design and its contributions are of potential interest to a wide audience. The proposed mechanisms are all simple yet effective, and the proofs are easy to follow. In particular, the construction of lower-bound results are quite interesting. On the other hand, the paper lacks any numerical examples/experiments to verify the effectiveness of their mechanisms in practice.

**Questions:**

- The mechanism design with hints framework studied in this paper is closely related to the ``Algorithms with predictions'' setup (see https://algorithms-with-predictions.github.io/ for a list of all papers on this topic). However, you haven't mentioned this in your related work. How does your result and contributions compare to the existing works in mechanism design with predictions? Are there any papers that are similar to yours?
- In this paper, you have focused on modular utilities, i.e., the utility of a subset of chosen buyers is simply the sum of their individual utilities. Have you thought about submodular utility functions? To what extent do your results hold if we move beyond modular utilities? What are the challenges with non-modular utilities?
- If the hints were potentially inaccurate, is it possible to design a mechanism that is robust (performs well even if the hints are wrong) and consistent (performs well when the hints are correct)? You have shown the consistency of your proposed mechanisms, are they robust as well?

**Reviewer Confidence:**

3: The reviewer is confident but not certain that the evaluation is correct

**Scope:**

4: The work is relevant to the Web and to the track, and is of broad interest to the community

---

### Official Review · Reviewer_yNkC · 2023-11-24

**Novelty:** 3
**Technical Quality:** 4

**Review:**

This paper studies the auction setting where a set of identical items is allocated to unit-demand agents and the mechanism does not have information on agents’ valuations (prior-free) other than that the welfare is within a range [alpha, 1]. The authors consider three different settings regarding feasible allocations:
- digital goods: the quantity of items is unbounded and can be allocated to any subset of agents.
- matroid setting: the agents are elements of a matroid and items can only be allocated to agents forming an independent set.
- public good setting: either everyone receives an item or no one receives an item.
The authors design a mechanism in each of the above three settings. For each mechanism, the authors analyze its competitive ratio – in terms of the number of agents and the parameter alpha – with respect to the optimal welfare. The authors show that the competitive ratio achieved by their mechanism is tight in both parameters in each of the three settings. All these results hold for alpha <= 0.5, and the regime for alpha > 0.5 is left for future work.

This paper is well-written and easy to follow. The setting studied in the paper is reasonable to me. The mechanisms are simple and natural. The competitive ratios are also tight.

However, I think the overall contribution of this paper is incremental. Technically, it seems to me that most of the analysis uses existing techniques. Conceptually, the main contribution to me is the new feature that the welfare is known to be within a certain range, which makes the model in between the completely prior-free model and the Bayesian model. This kind of feature already exists in other problems (e.g., online algorithms with predictions). Therefore, I also think the conceptual contribution of this paper is incremental.

**Questions:**

No questions.

**Ethics Review Description:**

N.A.

**Reviewer Confidence:**

2: The reviewer is willing to defend the evaluation, but it is likely that the reviewer did not understand parts of the paper

**Scope:**

3: The work is somewhat relevant to the Web and to the track, and is of narrow interest to a sub-community

---

### Decision · Program_Chairs · 2024-01-22

**Decision:**

Accept

**Comment:**

The paper studies the problem of designing prior-free revenue-maximizing mechanisms for allocating items to n buyers, where the mechanism has access to an estimate of the optimal welfare (guaranteed to be correct within some factor). They then develop mechanisms for three different settings.

 The reviewers are in agreement that the mechanisms are natural, the bounds are tight, and the proofs are simple (a feature, not a bug, in my opinion). There is some disagreement among the reviewers about whether the results are incremental; this, to me, is a fair question, but given the conciseness with which the results are presented, I lean towards evaluating the paper positively.